# The Role of GAB1 in Cancer

**DOI:** 10.3390/cancers15164179

**Published:** 2023-08-20

**Authors:** Manuel Jesús Pérez-Baena, Francisco Josué Cordero-Pérez, Jesús Pérez-Losada, Marina Holgado-Madruga

**Affiliations:** 1Instituto de Biología Molecular y Celular del Cáncer (IBMCC-CIC), Universidad de Salamanca/CSIC, 37007 Salamanca, Spain; mjperezbaena@usal.es (M.J.P.-B.); jperezlosada@usal.es (J.P.-L.); 2Instituto de Investigación Biosanitaria de Salamanca (IBSAL), 37007 Salamanca, Spain; 3Departamento de Medicina Interna, Complejo Asistencial de Zamora, 49022 Zamora, Spain; josue@usal.es; 4Departamento de Fisiología y Farmacología, Universidad de Salamanca, 37007 Salamanca, Spain; 5Instituto de Neurociencias de Castilla y León (INCyL), 37007 Salamanca, Spain; 6Virtual Institute for Good Health and Well Being (GLADE), European Campus of City Universities (EC2U), 86073 Poitiers, France

**Keywords:** GAB1, tumorigenesis, angiogenesis, metastasis, therapy resistance

## Abstract

**Simple Summary:**

GRB2-associated binder 1 (GAB1) is a docking protein pivotal in linking multiple stimuli to various intracellular signaling pathways. Embryonic development is disrupted in GAB1-deficient mice, and oncogenic mutations have been noted in cancer cases. In numerous types of cancer, high GAB1 expression levels correlate with a poor prognosis. Studies reveal that GAB1 significantly influences cellular transformation by changes in proliferation, evasion of apoptosis, metastasis, and angiogenesis—all fundamental processes in cancer development. Furthermore, GAB1 is implicated in the resistance/sensitivity to antitumor treatments, thus establishing its potential as an anticancer therapy target.

**Abstract:**

GRB2-associated binder 1 (GAB1) is the inaugural member of the GAB/DOS family of pleckstrin homology (PH) domain-containing proteins. Upon receiving various stimuli, GAB1 transitions from the cytoplasm to the membrane where it is phosphorylated by a range of kinases. This event recruits SH2 domain-containing proteins like SHP2, PI3K’s p85 subunit, CRK, and others, thereby activating distinct signaling pathways, including MAPK, PI3K/AKT, and JNK. GAB1-deficient embryos succumb in utero, presenting with developmental abnormalities in the heart, placenta, liver, skin, limb, and diaphragm myocytes. Oncogenic mutations have been identified in the context of cancer. GAB1 expression levels are disrupted in various tumors, and elevated levels in patients often portend a worse prognosis in multiple cancer types. This review focuses on GAB1’s influence on cellular transformation particularly in proliferation, evasion of apoptosis, metastasis, and angiogenesis—each of these processes being a cancer hallmark. GAB1 also modulates the resistance/sensitivity to antitumor therapies, making it a promising target for future anticancer strategies.

## 1. Introduction

GRB2-associated binding 1 (GAB1) is the inaugural member of the GAB family of docking proteins, which includes GAB1, GAB2, and GAB3 in mammals, Daughter Of Sevenless (DOS) in *Drosophila* and Suppressor Of Clear (Soc1) in *Caenorhabditis elegans* [1,2,3,4]. Historically, GAB1 was originally isolated as a binding partner of the SRC homology 3 (SH3) domain of growth factor receptor-bound protein 2 (GRB2) [5]; GAB2 was cloned as a partner of the SRC homology 2 (SH2) domain of the SHP2 phosphatase [6], and GAB3 was located by sequence similarity to GAB1 and GAB2 [7]. DOS was identified simultaneously by two independent research groups. The first one identified this protein as a substrate of the SHP2 orthologue in Drosophila and Corkscrew (Csw), and the second one by searching for mutants that suppress the rough-eye phenotype caused by a hyperactivated sevenless allele [8,9]. Soc1 was discovered from the search for suppressors of the fibroblast growth factor (FGF) orthologue in *C. elegans*, Egl-15 [10]. These proteins have a similar structural organization, despite sharing approximately 40–50% sequence similarity [5,11]. However, there seems to be a certain tissue tropism of expression of these proteins in mammals in spite of their ubiquitous expression. Thus, while GAB1 and GAB2 register their lowest expression levels in lymphoid tissues, GAB3 reaches its highest expression values [4]. The *GAB1* gene is located on chromosome 4q31.1 in humans [12]. Two isoforms of GAB1, lacking catalytic activity, have been described, one high molecular weight (120 kDa) expressed exclusively in the heart and one low molecular weight (100 kDa) expressed ubiquitously [4,11,13]. However, the role of different GAB1 isoforms in different cellular contexts and under different stimuli remains unknown. Like its homologues, GAB1 has a highly conserved pleckstrin homology (PH) domain (amino acids 14–99) at its amino-terminal end involved in the interaction with phosphatidylinositol 3,4,5-triphosphate (PIP3) present in the plasma membrane, a central proline-rich domain involved in the interaction with proteins with SRC homology 3 (SH3) domains, 47 predicted phosphorylation sites on serine/threonine residues, and 16 potential phosphotyrosine sites involved in the recruitment of proteins with SRC homology 2 (SH2) domains [5,11]. Unlike its other family members, GAB1 has a MET-binding domain (MBD) (amino acids 450–532) within the proline-rich domain, responsible for direct association with the MET receptor [14]. In addition, GAB1 has a nuclear localization signal (amino acids 15–23) at its amino-terminal end [15] (Figure 1). GAB1-deficient embryos succumb in utero, presenting with developmental abnormalities in the heart, placenta, liver, skin, limb, and diaphragm myocytes [16,17].

## 2. Participation of GAB1 in Cell Signal Transduction

GAB1 is phosphorylated at tyrosine residues in response to a wide variety of stimuli including epidermal growth factor (EGF), platelet-derived growth factor (PDGF), nerve growth factor (NGF), hepatocyte growth factor (HGF), vascular endothelial growth factor (VEGF), keratinocyte growth factor (KGF), fibroblast growth factor (FGF), and insulin and stem cell factor (SCF); cytokines such as erythropoietin (EPO), interleukin-6 (IL-6), interferon-α (IFN-α), interferon-γ (IFN-γ), interleukin-3 (IL-3), and thrombopoietin (TPO); and various immune stimuli such as anti-IgM or F(ab’)2 and lysophosphatidic acid (LPA) [11]. These phosphotyrosine residues are recognized by different proteins with SH2 domains, such as the SH2-containing protein tyrosine phosphatase-2 (SHP2) (Y627, Y659), p85 subunit of phosphoinositide 3-kinase (PI3K) (Y447, Y472, Y589), chicken tumor virus number 10 (CT10) regulator of kinase (CRK) (Y424, Y259, Y317), and others triggering the activation of different signaling pathways [4,14,18]. Furthermore, GAB1 is not only phosphorylated by receptor protein tyrosine kinases, but also by kinases of the SRC family [19,20]. An approximately 40% reduction in GAB1 phosphorylation levels was observed following HGF stimulation in cells derived from SRC null mice. Furthermore, the expression of a constitutively active mutant of SRC increased HGF-independent GAB1 phosphorylation. Moreover, SRC has been shown to interact directly and phosphorylate *GAB1* through its SH2 domains by in vitro kinase assays [19]. Similarly, hematopoietic cell kinase (HCK) has been reported to induce GAB1 phosphorylation in response to IL-6 in multiple myeloma cells [20]. It has also been shown that GAB1 can be phosphorylated by the non-receptor tyrosine kinase malectin-like receptor kinase FERONIA (FER) at tyrosine residue 627 [21].

Analogously, GAB1 is also phosphorylated in response to different stresses such as oxidative stress, fluid shear stress and ultraviolet (UV) light [22,23,24]. It has been outlined that hydrogen peroxide stimulation induces a dose- and time-dependent phosphorylation of GAB1. Furthermore, GAB1 has been shown to be an integrator of cell death through its interaction with SHP2- jun N-terminal kinase (JNK) activation versus cell survival mediated by p85/PI3K/ thymoma viral oncogene homolog (AKT) pathway activity in oxidative stress [24]. By comparison, fluid shear stress-induced endothelial nitric oxide synthase (eNOS) activation has been found to be GAB1-dependent [23,25]. Ultraviolet radiation has been described to induce activation of the JNK signaling pathway to trigger cell apoptosis. In this context, it has been established that GAB1-deficient mouse fibroblast cells do not induce JNK activation as measured by UV, as well as a partial induction of caspase 3, DNA fragmentation, and YOPRO-1 staining [22].

GAB1 recruitment from the cytoplasm to activated membrane receptors can be direct or indirect (Figure 2). The direct recruitment mode (Figure 2A) appears to be unique to the c-MET receptor. The GAB1 MBD domain interacts directly with the activated kinase domain of c-MET upon binding to its ligand HGF, allowing signal transduction [26,27,28]. On the other hand, the c-MET receptor also indirectly recruits GAB1 via growth factor receptor-bound protein 2 (GRB2) or via SHC-GRB2 [27,29]. Upon the tyrosine kinase receptor activation, the phosphotyrosine residues present in its cytoplasmic tails are recognized by the SH2 and/or phosphotyrosine-binding (PTB) domains of GRB2, and GRB2 then binds to the proline-rich region of GAB1 via its C-terminal SH3 domain [4,30]. The interaction of GRB2 with GAB1 is mediated by distinct domains: the canonical proline-rich GRB2-binding domain, PX3RX2KPX7PLD, and the atypical GRB2-binding domain present in the MBD, PXXXR [31]. This indirect mode of GAB1 recruitment to activated receptors seems to apply to all other receptors that recruit GAB1 (Figure 2B).

As mentioned, GAB1 can channel signaling from the external stimulus to the PI3K/AKT pathway. Activation of this pathway is triggered by the direct interaction of GAB1 with the SH2 domain of the p85 regulatory subunit of PI3K kinase [32,33]. Mutations experiments on the binding sites of GAB1 to p85 have been reported, resulting in the loss of signal transduction under several stimuli [32,34,35]. It has been demonstrated that this interaction is essential for eyelid closure and for keratinocyte migration through the generation of knock-in mice expressing a GAB1 protein defective in p85 recruitment [27]. The GAB1-p85 regulatory subunit of PI3K interaction has been described as critical for the flow-stimulated PI3K/AKT/eNOS signaling pathway in endothelial cells [23]. GAB1 overexpression enhances AKT activation induced by FGF, VEGF, and HGF, whereas overexpression of the p85-binding site mutant version of GAB1 results in decreased AKT activation [36]. This mutant also fails to produce NGF-induced antiapoptotic signal transmission [32]. Furthermore, PI3K phosphorylates phosphatidylinositol-3,4-diphosphate (PIP2) to phosphatidylinositol-3,4,5-triphosphate (PIP3) [37,38,39]. This PIP3 recruits GAB1 to the plasma membrane via the PH domain, leading to further activation of PI3K. This positive feedback loop amplifies signals from different growth factors through this pathway [33,40,41].

Among other proteins that bind to GAB1, the role of the SHP2 protein, which interacts with the phosphotyrosine residues Y627 and Y659 located at the GAB1 C-terminal end, is noteworthy [42]. The GAB1–SHP2 interaction is possibly the best characterized. All members of the GAB family have been shown to interact with SHP2 or its homologues [6,7,8,11]. This finding indicates that interaction with SHP2 is a conserved feature of the different members of the GAB protein family. The functional significance of this interaction has been studied using mutants of GAB1 that are unable to bind SHP2. This mutant has been reported to cause alterations in MET-dependent morphogenesis and block the activation of the mitogen-activated protein kinase (MAPK) signaling pathway mediated by HGF, EGF, and LPA [42,43,44,45]. In addition, the Y627 tyrosine residue of GAB1 that enables GAB1–SHP2 interaction is essential for protein kinase A (PKA)-dependent fluid shear stress-induced eNOS activation [25]. GAB1 also participates in the maintenance of MAPK activation downstream of IL-6 signaling [46]. Furthermore, the generation of knock-in mice expressing a GAB1 mutant lacking the SHP2 interaction site has shown that the GAB1–SHP2 complex is essential for Ras/ extracellular signal-regulated kinase 1/2 (ERK1/2) pathway activation. These mice show abnormal placenta, an organ dependent on ERK signaling [47,48]. Surprisingly, this recruitment is also required for SHP2 accessibility to its substrates. However, the mechanism by which this interaction is required and the substrates of SHP2 are not yet well elucidated [11,42,49].

As described above, GAB1 is phosphorylated in response to EGF and is involved in PI3K/AKT and MAPK pathway signaling. GAB1 has been shown to be strictly required in EGF-induced activation of the PI3K/AKT signaling pathway by associating it with the p85 subunit of PI3K [33,50]. Furthermore, GAB1 overexpression has been described to potentiate EGF-induced activation of the MAPK pathway [33]. On the other hand, it has been reported that GAB1 downregulation in human head and neck squamous cell carcinoma cell lines (HN4, HN6, HN12, HN13, and HN31) reduced PI3K/AKT and MAPK pathway-mediated signaling as well as the duration of signaling after EGF stimulation, potentially affecting EGFR stability [51]. Therefore, these studies demonstrate that GAB1 exerts an important role in the amplification and maintenance of EGF-induced PI3K/AKT and MAPK signaling.

On the other hand, GAB1 has been described to contain potential binding sites for the SH2 domain of phospholipase C γ (PLCγ), CRK, CRK-like (CRKL), and other proteins such as P21-activated kinase 4 (PAK4) or some members of the partitioning defective (PAR) complex [52,53,54,55]. Phosphorylation of GAB1 at tyrosine residues Y307, Y373, and Y407 has been shown to generate a binding site for PLCγ, which is required for HGF-mediated tubulogenesis [56]. Upon HGF stimulation, the GAB1–CRK interaction leads to RAC activation, enhancing cell scattering, invasive capacity, and xenograft growth in synovial sarcoma cell lines [53]. GAB1 has been found to interact with CRKL by generating a GAB1 mutant version with an internal amino acid deletion of 242–410. Furthermore, this interaction is required for HGF-mediated activation of the repressor/activator protein 1 (RAP1) GTPase [54]. Similarly, the GAB1–PAK4 complex cooperates in HGF-induced epithelial cell scattering and invasiveness [52]. Recently, GAB1 has been reported to act as a negative regulator of cell polarity through interaction with the various proteins of the PAR complex, particularly PAR1 and PAR3 [55].

In addition to the functions of GAB1 as a molecular integrator of different signals, GAB1 also performs other functions. GAB1 acts as a molecular switch that allows the interconnection of different cell signaling pathways such as the connection between the EGFR/PI3K/AKT pathway and the nuclear factor kappa-light-chain-enhancer of activated B cells (NF-kB) pathway in glioblastoma cells. This interconnection has been determined to be exerted through the GAB1–SHP2 complex using the mutant version of GAB1 unable to bind SHP2 (Y627F). Furthermore, in this interconnected system of cell signaling pathways, the phosphatase domain of SHP2 exerts a negative regulatory effect on GAB1 phosphorylation and, therefore, on NF-kB activity [57]. Additionally, in the context of urothelial cell carcinoma, it has been described that GAB1 interacts with mammalian target of rapamycin complex (mTORC) and activates it after exposure to EGF. Specifically, it has been depicted that the recruitment of mTORC to the plasma membrane and, therefore, the regulation of its activity is mediated through the PH domain of GAB1 [58]. Similarly, it has been found that ERK1/2 moves to the nucleus by interacting with GAB1 through the MET-binding domain, in which a nuclear localization signal is found [15]. These data suggest that GAB1 also behaves as a relocalizing element of different proteins.

All these findings point to GAB1 as a platform for integrating external signals to trigger and amplify their emergent signaling. Consequently, these results suggest the presence of sophisticated regulatory mechanisms. Since the PH domain of GAB1 is involved in the interaction with PIP3 present in biological membranes and these are products of PI3K kinase action, it has been reported that the lipid phosphatase and tensin homologue (PTEN) or SH2 domain-containing inositol 5-phosphatases 1/2 (SHIP1/2) could negatively influence the GAB1 membrane localization and thus lead to its inactivation [59,60]. Furthermore, SHP2 has been shown to dephosphorylate the tyrosine residues of GAB1 involved in the recruitment of p85 and Ras GTPase-activating protein (RasGAP) [61,62]. On the other hand, a mechanism of GAB1 activity regulation by ERK1/2-mediated phosphorylation has been described, although its functional consequences can be both positive and negative in a context-dependent manner [63,64,65]. Upon HGF stimulation, GAB1 is phosphorylated by ERK at threonine residue 477, enhancing the interaction of p85 to GAB1. Considering that GAB1 regulates MAPK pathway activation via SHP2 and that the PI3K/AKT signaling pathway activates ERK, this regulatory system could result in a positive feedback loop of the pathway [66]. Conversely, EGF stimulation induces ERK2-mediated phosphorylation of GAB1 at other residues that inhibit cell signaling via the PI3K/AKT pathway [63]. It will be essential to determine how and when ERK2 regulates the channeling of signaling of different stimuli through GAB1. Additionally, in the case of MET signaling, PKC-α- and PKC-β1-mediated phosphorylation has been found to negatively regulate GAB1 activity [65].

## 3. Role of GAB1 in Cancer

GAB1 is involved in diverse biological processes such as cell proliferation, survival, invasion and migration, cell differentiation, angiogenesis, and inflammation [11,67,68,69,70]. All these functions of GAB1 appear to be fundamental to the processes underlying malignant transformation, including tumor angiogenesis and metastasis (Figure 3). GAB1 expression appears to be deregulated in a wide variety of tumors such as thyroid cancer, cervical cancer, breast cancer, meningiomas, cholangiocarcinoma, medulloblastomas, chronic lymphocytic leukemia, head and neck cancer, and colorectal cancer, among others (Figure 4). In breast cancer, GAB1 cancer-associated mutations, Y83C and T387N, have been described and characterized. These GAB1 mutant versions resulted in the acquisition of a more elongated fibroblastic phenotype in immortalized MCF10A mammary epithelial cells, EGF-independent proliferation, and increased ERK activation [71,72]. In addition, genetic rearrangements involving GAB1 have been described in different pediatric and adult tumor contexts. GAB1:ABL1 (*Abelson*) fusion has been reported in perineurioma, angiofibroma, and solitary fibrous tumors [73]. Also, eukaryotic translation initiation factor 4 gamma 2 (EIF4G2)–GAB1 fusion has been found in patients with non-small cell lung cancer treated with EGFR TKI [74]. GAB1 germline copy number variation is linked to breast cancer risk [75]. Furthermore, genetic polymorphisms in GAB1 have been reported to increase susceptibility to developing some types of cancer, such as cholangiocarcinoma, meningiomas, and lung cancer [76,77,78,79]. Interestingly, high levels of GAB1 expression are associated with poor prognosis in patients with gliomas, hepatocellular carcinoma, and ovarian cancer [80,81,82]. It has been delineated that GAB1 is a biomarker of poor prognosis in meningiomas, medulloblastomas, and bone and soft tissue sarcomas [83,84,85,86].

This review focuses on GAB1’s influence on cellular transformation particularly in proliferation, evasion of apoptosis, metastasis, and angiogenesis—each of these processes being a cancer hallmark. GAB1 also modulates the resistance/sensitivity to antitumor therapies.

### 3.1. GAB1 in Proliferation and Apoptosis

Alteration of cell cycle control and apoptosis evasion are two hallmarks of cancer. In addition, another typical feature of tumor cells is uncontrolled proliferation [87,88]. It has been shown that cell cycle progression induced by the expression of the MET receptor oncoprotein (Tpr-Met) in *Xenopus laevis* oocytes depends on GAB1. The Met-binding domain, the pleckstrin homology domain, and the PI3K and SHP2 binding sites are required to perform this function. Furthermore, this GAB1-dependent regulation of cell cycle progression is also observed in the case of the FGF receptor [89]. Suppression of GAB1 expression leads to G1 arrest in hilar cholangiocarcinoma and chondrosarcoma cell lines [90,91]. Also, GAB1 expression lacking the PH domain is associated with neoplastic progression by enhancing colony-forming ability [92]. GAB1 downregulation inhibits proliferation in hilar cholangiocarcinoma cell lines by decreasing PI3K/AKT signaling pathway activation [90]. GAB1 overexpression promoted cell proliferation in oral squamous carcinoma cells by activating the AKT/CDH1 pathway, and its silencing promoted cell apoptosis [93]. Moreover, it has been found that microRNA-29a-3p downregulation induces GAB1 upregulation to promote glioma cell proliferation [94]. In hepatocellular carcinoma cell lines, GAB1 knockdown inhibited cell proliferation by reducing of ERK1/2 activation [95]. The absence of protein tyrosine kinase 6 (PTK6) expression reduces the proliferative capacity of cervical cancer cells and apoptosis induction in a GAB1-dependent manner [96]. HCK-mediated phosphorylation of GAB1 induces proliferation and survival in IL-6-induced multiple myeloma cells. Furthermore, IL-6 induces the association of GAB1 with SHP2 and CRKL in these tumor cells, promoting cell proliferation [20].

Fibroblasts’ oncogenic transformation leads to morphological changes that allow uncontrolled proliferation and progression even in the presence of inhibitory signals such as cell–cell contact. The Tpr-Met and epidermal growth factor receptor 2 (EGFR2, ErbB-2, Neu) oncoprotein has been reported to induce oncogenic transformation, and this mechanism has been demonstrated to require GAB1 involvement [89,97,98]. On the other hand, GAB1 has been found to enhance cell growth and induce soft agar colony formation in NIH3T3 cells and this is dependent on interaction with SHP2 [5,99]. Moreover, the Py772- EPH Receptor A2 (EphA2) oncoprotein has been described to induce proliferation and anchorage-independent growth and tumorigenicity in vivo in nasopharyngeal carcinoma cell lines in a GAB1-dependent manner [100].

GAB1 suppression increased apoptosis in hilar cholangiocarcinoma, chondrosarcoma cell lines by modulating the B-cell lymphoma 2 (BCL-2)/ Bcl-2 Associated X-protein (BAX) axis, and pancreatic carcinoma through microRNA-383 regulation. It has been also depicted that GAB1 leads to partial apoptosis induction in HCT116 colorectal cancer cells through miR-5582-5p modulation [90,91,101,102]. 

### 3.2. GAB1 in Angiogenesis

Angiogenesis is a complex multistep process in which new blood vessels are produced from pre-existing ones. This complex biological process is orchestrated by the interaction of a wide variety of mediators, as well as the involvement of different cell types [103]. In 1971, Folkman demonstrated that angiogenesis was crucial for the development and growth of solid tumors beyond 1–2 mm^3^ [104,105]. Therefore, angiogenesis is currently considered a hallmark of cancer [87,88]. Following this discovery, the development of antiangiogenic therapies has become of great interest for study [104]. Tumor angiogenesis occurs due to an imbalance between proangiogenic and antiangiogenic mediators. This alteration may be due to genetic mutations in genes that control the production of angiogenic regulators and to different types of stresses such as hypoxia, acid pH, and hypoglycemia [106]. Among all these types of metabolic stresses, the best studied is tumor hypoxia, which markedly drives tumor angiogenesis by inducing the production of VEGF and other angiogenesis-positive agents [107].

Tumors engrafted in GAB1-ecKO mice have been shown to have a substantially lower level of capillary density, as well as a marked decrease in tumor weight and volume [108]. It has been reported that GAB1 controls the autocrine secretion of VEGF in hilar cholangiocarcinoma tumor cells, promoting angiogenesis and tumor invasion [109]. Moreover, GAB1 is associated with HGF-stimulated VEGF production in EGFR-mutant lung cancer cell lines [110].

The role of GAB1 in the promotion and control of physiological angiogenesis is well known [111]. However, further studies on the role of GAB1 in tumor angiogenesis are needed. In addition, it might be useful to study the role of this protein in the response to antiangiogenic therapies.

### 3.3. GAB1 in Tumor Migration and Invasion

The presence of metastases is often responsible for high cancer mortality [112]. Tumor cells can metastasize to sites adjacent to or distant from the primary site due to their capacity for cell migration and invasion [113]. GAB1 downregulation reduced migratory and invasive capacity in hepatocellular carcinoma, hilar cholangiocarcinoma, oral squamous cell carcinoma, and colorectal carcinoma cell lines [90,93,95,114,115]. In hepatocellular and colorectal carcinomas, the association of reduced migratory and invasive capacity related to GAB1 deletion is attributed to microRNA-200A, microRNA-105, and microRNA-409-3p, respectively [95,114,115]. In intrahepatic cholangiocarcinoma, GAB1 regulates migration through the PI3K/AKT signaling pathway [116]. Furthermore, GAB1 has been described to be upregulated to promote migration in anaplastic thyroid carcinoma cells through modulation of the AKT–Multidrug Resistance Protein 1 (MDR1) axis [117]. GAB1 overexpression increases migratory and invasive activity in cervical cancer cell lines [96]. Also, GAB1 has been reported to be closely involved in glioblastoma cell invasion through HGF-induced activation of dedicator of cytokinesis 7 (DOCK7) and RAC1 [118]. Likewise, GAB1 has been found to increase the migration of chronic lymphocytic leukemia cells by forkhead box 1 (FOXO1) regulation [119]. Furthermore, in breast cancer, GAB1 overexpression has been described to promote metastasis in vivo by PAR complex dissociation [120]. On the other hand, GAB1 recruitment and phosphorylation for activation of the SHP2-ERK pathway have been shown to be essential in enhancing FER-induced metastasis in ovarian cancer [21].

Epithelial–mesenchymal transition (EMT) is the process by which epithelial cells lose polarity and adhesion to acquire mesenchymal cell characteristics. The acquisition of these mesenchymal features depends on a wide variety of intracellular signaling networks such as the MAPK pathway, PI3K/AKT, and others. In relation to EMT, the final convergence of these cell signaling pathways is the expression of EMT-inducing transcription factors such as SNAIL, SLUG, XEB1, and TWIST, among others. This process is the underlying mechanism for the development of tumor metastasis [121,122]. High levels of GAB1 expression have been described in metastatic HER2+ and triple-negative breast cancer patients. In vitro, elevated GAB1 expression has been reported to enhance the migration of MDA-MB231 and SK-BR3 breast cancer cells through dissociation of the polarity-associated partitioning defective (PAR) complex. It is worth noting that cells must lose cell polarity to migrate. GAB1 overexpression in these cell lines led to decreased expression levels of E-cadherin and Zonula occludens 1 (ZO-1) with increased expression levels of N-cadherin and vimentin [120]. Additionally, CRK knockdown in HGF-stimulated bladder cancer cells has been reported to induce E-cadherin expression, downregulation of epithelial–mesenchymal transition markers such as N-cadherin and vimentin, and decreased levels of GAB1 phosphorylation. Thus, CRK has been shown to play a pivotal role in the induction of HGF-mediated EMT through sustained phosphorylation of GAB1 [123]. In addition, the opioid receptor Mu has been illustrated to promote EGF-induced migration and mesenchymal epithelial–mesenchymal transition through GAB1 recruitment in the lung cancer cell line H358 [124]. Moreover, EGF increases transforming growth factor-β (TGF-β)-induced EMT in lung and pancreatic cancer cells and is due to the promotion of SHP2 binding to GAB1 [125]. These results support a positive role for GAB1 in promoting cell migration through the EMT process.

Recent studies indicate that different elements of the tumor microenvironment such as the extracellular matrix, fibroblasts, endothelial cells, and different immune populations can modulate the migratory and invasive capabilities of cancer cells [126]. In most tumors, tumor-associated macrophages (TAMs) represent the main stromal immune population. These TAMs produce and release several inflammatory components that play a positive role in cancer metastasis [127]. GAB1 has been reported to be required for the stimulation of macrophage-mediated invasion in gastric carcinoma cell line CR-1739 and colorectal carcinoma cell line CRL-2577 [128]. This study provides a solid basis for investigating the role of GAB1 in the tumor microenvironment.

### 3.4. GAB1 in Resistance to Tumor Therapy

The clinical efficacy of cancer treatment has increased thanks to the contribution of numerous studies and the increasing discovery of new targeted therapies. However, cancer is a very heterogeneous and dynamic disease, so as treatment progresses, treatment efficacy may be lost or even lead to a complete loss of therapeutic response following the emergence of resistance mechanisms [129]. 

GAB1 has been described to be involved in alectinib resistance mediated by activation of the HGF/MET signaling pathway in anaplastic lymphoma kinase (ALK)-positive non-small cell lung cancer [130]. It has also been shown that GAB1 participates in the mechanism of resistance to fibroblast growth factor receptor (FGFR) inhibitors, PD173074 and BGJ398, through MET-induced activation of AKT and ERK in FGFR1-amplified lung cell lines [131]. Additionally, GAB1 has been found to exert a positive role in HGF-mediated resistance to the V-Raf murine sarcoma viral oncogene homologue B (BRAF) kinase inhibitors, dabrafenib, and trametinib, and to the mitogen-activated protein kinase kinase (MEK) inhibitor, PD0325901, in BRAFV600E mutant melanoma [132]. GAB1 has been depicted as participating in the mechanism of HGF-induced EGFR-tyrosine kinase inhibitors (TKIs, osimertinib and gefitinib) resistance in lung cancer cell lines [110,133]. GAB1 has been established to participate in HGF-induced resistance to cetuximab in lung cancer cell lines both in vitro and in vivo by promoting cell survival through AKT activation [134]. Also, head and neck squamous carcinoma cells are highly dependent on GAB1 in the presence of the SHP2 inhibitor SHP099 [135]. Additionally, it has been described that decreased GAB1 expression levels in head and neck cancer cell lines (HN4, HN6, HN12, HN13, and HN31) increased sensitivity to the EGFR inhibitor gefitinib by decreasing MAPK and AKT phosphorylation [51]. On the other hand, GAB1 has been reported to be responsible for restoring PI3K/AKT1 pathway activation in HGF-induced resistance to MET inhibitors (JNJ-38877605, PHA-665752, crizotinib) and anti-MET antibody (DN30 Fab) [136]. GAB1 is associated with tamoxifen resistance in metastatic breast cancer patients [137]. Furthermore, it has been reported that chronic lymphocytic leukemia (CLL) cells treated with ibrutinib lead to an increase in *GAB1* mRNA and protein levels. Elevated GAB1 expression levels in ibrutinib-treated CLL cells have been associated with augmented tonic activation of pAKT, implying cellular relocalization signals and cell survival. Thus, GAB1 inhibitors reduce cell migration and increase cell apoptosis individually or in combination with ibrutinib by altering the FOXO1–GAB1–pAKT axis [119]. On the other hand, numerous investigations have studied IL-6-triggered dexamethasone resistance in multiple myeloma. In addition to other resistance mechanisms, using GAB1 and SHP2-GAB1 binding mutants, GAB1 has been shown to be associated with the IL-6-induced dexamethasone resistance mechanism in multiple myeloma cells [138].

Besides the use of chemotherapy as a treatment for cancer patients, radiotherapy is also an effective therapeutic option to treat some tumor types [139]. GAB1 is involved in MET-mediated radiotherapy resistance mechanisms, promoting invasive growth in breast cancer, melanoma, and glioblastoma cell lines [140].

### 3.5. GAB1 Inhibitors

As mentioned above, GAB1 acts as a signal integrator of multiple signaling pathways and is deregulated in many cancers. Considering the GAB1 involvement in the different hallmarks of cancer, this protein could be a potential therapeutic target to treat different tumor types. Recently, selective inhibitors of GAB1 have been developed. These new compounds, GAB-001 and GAB-004, target the PH domain of GAB1 by inducing a conformational change that prevents its phosphorylation. They have only been tested in breast cancer cell lines MDA-MB231 and T47D, showing potent cytotoxicity [141]. However, much future work and further experimental validation are needed to develop new compounds for translation to clinical practice. The main limitation in developing selective inhibitors against this protein is based on the absence of its protein structure [141].

## 4. Conclusions

GAB1 behaves as a protein integrator of multiple cell signaling pathways regulating cell survival and proliferation, angiogenesis, and cell migration and invasion, facilitating cellular transformation, thus playing a crucial role in tumorigenesis. GAB1 also mediates resistance/sensitivity to various chemotherapeutic drugs and radiotherapy. GAB1 expression is altered in a wide variety of tumors and is associated with poor patient prognosis in several tumor types. GAB1 inhibitors have a good prospect in cancer therapy. Therefore, GAB1 could be considered a potential therapeutic target for cancer treatment. However, further pre-clinical studies and clinical trials are needed for the clinical application of GAB1 inhibitors in specific tumor contexts.

## Figures and Tables

**Figure 1 cancers-15-04179-f001:**
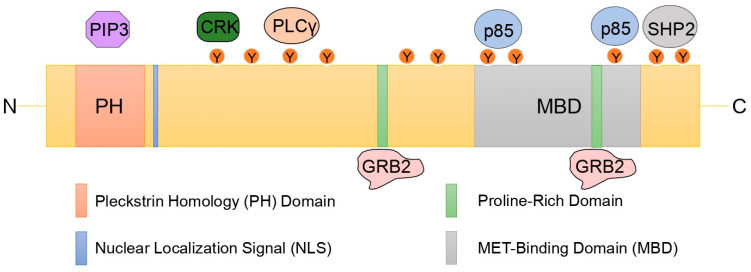
GAB1 structure and partner binding sites. Schematic representation of the GAB1 structure. GAB1 has a pleckstrin homology (PH) domain at its amino-terminal end involved in the interaction with phosphatidylinositol 3,4,5-triphosphate (PIP3), a nuclear localization signal (NLS), tyrosine phosphorylation residues involved in the interaction with different proteins such as CRK, PLCγ, p85, and SHP2, a proline-rich region involved in the canonical and atypical interaction with GRB2, and the MET-binding domain (MBD) involved in the interaction with MET receptor.

**Figure 2 cancers-15-04179-f002:**
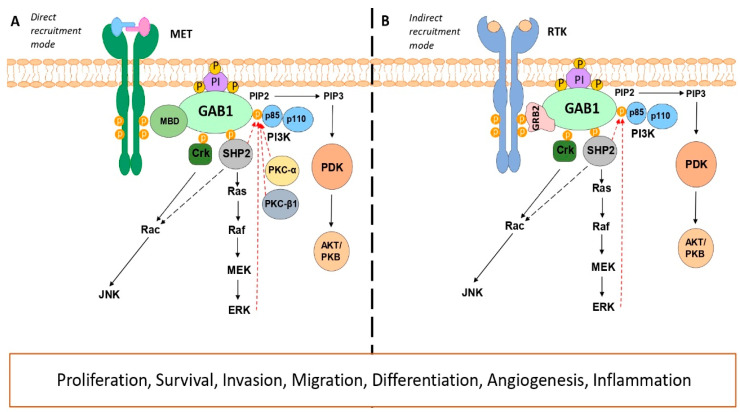
GAB1 recruitment modes to activated receptors and major GAB1 signaling pathways. The direct mode of recruitment appears to be exclusive to the c-MET receptor (**A**), although it can also recruit GAB1 indirectly, like the other tyrosine kinase receptors (**B**). Physical interaction between GAB1 and CRK, SHP2, and p85 has been described, triggering the activation of the JNK, Ras/MAPK, and PI3K/AKT cell signaling pathways, respectively. ERK positively or negatively regulates GAB1 phosphorylation in a context-dependent manner, while SHP2 negatively regulates it. In addition, in response to HGF, PKC-α and PKC-β1 regulate GAB1 activation in a negative way. Red arrows denote negative regulation.

**Figure 3 cancers-15-04179-f003:**
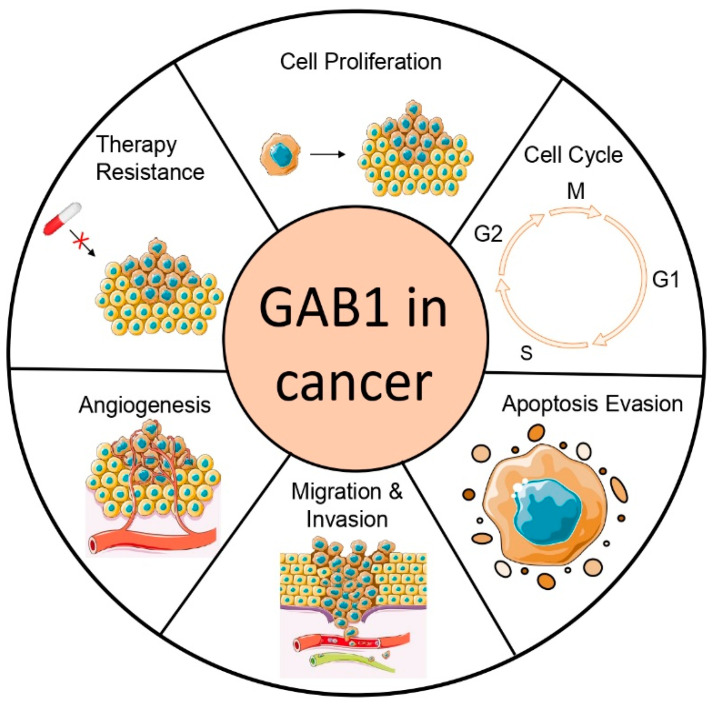
GAB1 is involved in cancer development and evolution. The GAB1 signaling pathway plays an important role in cancer pathobiology, including tumor cell proliferation, cell cycle, apoptosis evasion, tumor cell invasion and migration, tumor angiogenesis, and therapy resistance. Parts of the figure were drawn using pictures from Servier Medical Art. Servier Medical Art by Servier is licensed under a Creative Commons Attribution 3.0 Unported License (https://creativecommons.org/licenses/by/3.0/ accessed on 14 July 2023).

**Figure 4 cancers-15-04179-f004:**
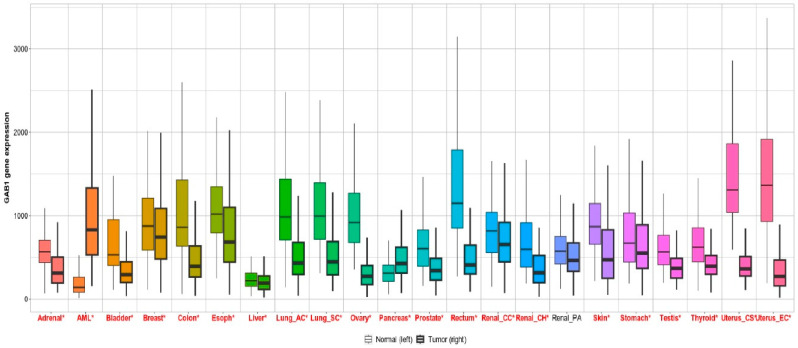
Boxplot of *GAB1* mRNA level in normal and tumor human tissues. Bulk tissue gene expression for *GAB1* from TNMplot (https://tnmplot.com/analysis/ (accessed on 14 July 2023)). Significant differences by the Mann–Whitney U test are marked with * and red legend.

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
