# Peer review of "The Role of GAB1 in Cancer"

_cancers, 2023, doi:10.3390/cancers15164179_

Round 1

Reviewer 1 Report

This is a concise and well done review that appropriately highlights the role of an increasingly important docking protein (GAB1) in various aspects of cancer biology. The paper is to the point, nicely referenced and should be of interest to cancer biologists and researchers in the signal transduction field.

Author Response

Reviewer: 1

This is a concise and well done review that appropriately highlights the role of an increasingly important docking protein (GAB1) in various aspects of cancer biology. -The paper is to the point, nicely referenced and should be of interest to cancer biologists and researchers in the signal transduction field.

ANSWER: Many thanks for your kind comments.

Reviewer 2 Report

In this review, the authors present an overview of GAB1 in cancer.

 Overall the review is very interesting in many scientific aspects, I thereby would to include additional

study for this manuscript.  I would like the authors to include these studies also in the manuscript so

that increase the strengthening of the overview of GAB1 in  cancer.

MAPK and AKT pathway play a important role in cancer and how GAB1 in supporting MAPK and PI3K

signaling through EGFR receptor. It is also known that the downregulation of GAB1 increases EGFR

inhibitor sensitivity.

The following study can be included in the manuscript.

Hoeben at al, have shown previously demonstrating the importance of GAB1 in supporting MAPK and  

PI3K signaling, and showed that this increased sensitivity to EGFR inhibition, and reduced EGFR signaling

through the AKT and MAPK pathways, potentially through affecting EGFR stability (PMID: 22865653).

Based  on these studies, it would be very important to analyze the EGFR pathway in greater detail,

and evaluating levels of EGFR stability induced by GAB1. It is also important to benchmark signaling

results to use of an EGFR inhibitor.

The review studies documented in this manuscript are very interesting and also can include the

following studies above for strengthening the manuscript.

Author Response

Reviewer: 2

In this review, the authors present an overview of GAB1 in cancer.

Overall, the review is very interesting in many scientific aspects, I thereby would to include additional study for this manuscript.  I would like the authors to include these studies also in the manuscript so that increase the strengthening of the overview of GAB1 in cancer.

MAPK and AKT pathway play an important role in cancer and how GAB1 in supporting MAPK and PI3K signaling through EGFR receptor. It is also known that the downregulation of GAB1 increases EGFR inhibitor sensitivity.

The following study can be included in the manuscript.

Hoeben at al, have shown previously demonstrating the importance of GAB1 in supporting MAPK and PI3K signaling, and showed that this increased sensitivity to EGFR inhibition, and reduced EGFR signaling through the AKT and MAPK pathways, potentially through affecting EGFR stability (PMID: 22865653).

Based on these studies, it would be very important to analyze the EGFR pathway in greater detail, and evaluating levels of EGFR stability induced by GAB1. It is also important to benchmark signaling results to use of an EGFR inhibitor.

The review studies documented in this manuscript are very interesting and also can include the following studies above for strengthening the manuscript.

ANSWER: Many thanks for your comments and suggestions, which have helped greatly improving our manuscript.

Page 05, Lines 179-189. “As described above, GAB1 is phosphorylated in response to EGF and is involved in PI3K/AKT and MAPK pathway signaling. GAB1 has been shown to be strictly required in EGF-induced activation of the PI3K/AKT signaling pathway by associating it with the p85 subunit of PI3K [33,50]. Furthermore, GAB1 overexpression has been described to potentiate EGF-induced activation of the MAPK pathway [33]. On the other hand, it has been reported that GAB1 downregulation in human head and neck squamous cell carcinoma cell lines (HN4, HN6, HN12, HN13, and HN31) reduced PI3K/AKT and MAPK pathway-mediated signaling as well as the duration of signaling after EGF stimulation, potentially affecting EGFR stability [51]. Therefore, these studies demonstrate that GAB1 exerts an important role in the amplification and maintenance of EGF-induced PI3K/AKT and MAPK signaling.”

Page 10, Lines 424-427. “Besides, it has been described that decreased GAB1 expression levels in head and neck cancer cell lines (HN4, HN6, HN12, HN13, and HN31) increased sensitivity to the EGFR inhibitor gefitinib by decreasing MAPK and AKT phosphorylation [51].”
